# The Pattern and Function of DNA Methylation in Fungal Plant Pathogens

**DOI:** 10.3390/microorganisms8020227

**Published:** 2020-02-08

**Authors:** Chang He, Zhanquan Zhang, Boqiang Li, Shiping Tian

**Affiliations:** 1Key Laboratory of Plant Resources, Institute of Botany, Chinese Academy of Sciences, Beijing 100093, China; changhe@ibcas.ac.cn (C.H.); zhangzhanquan82@ibcas.ac.cn (Z.Z.); bqli@ibcas.ac.cn (B.L.); 2College of Life Sciences, University of Chinese Academy of Sciences, Beijing 100049, China

**Keywords:** DNA methylation, fungal plant pathogen, development, pathogenicity

## Abstract

To successfully infect plants and trigger disease, fungal plant pathogens use various strategies that are dependent on characteristics of their biology and genomes. Although pathogenic fungi are different from animals and plants in the genomic heritability, sequence feature, and epigenetic modification, an increasing number of phytopathogenic fungi have been demonstrated to share DNA methyltransferases (MTases) responsible for DNA methylation with animals and plants. Fungal plant pathogens predominantly possess four types of DNA MTase homologs, including DIM-2, DNMT1, DNMT5, and RID. Numerous studies have indicated that DNA methylation in phytopathogenic fungi mainly distributes in transposable elements (TEs), gene promoter regions, and the repetitive DNA sequences. As an important and heritable epigenetic modification, DNA methylation is associated with silencing of gene expression and transposon, and it is responsible for a wide range of biological phenomena in fungi. This review highlights the relevant reports and insights into the important roles of DNA methylation in the modulation of development, pathogenicity, and secondary metabolism of fungal plant pathogens. Recent evidences prove that there are massive links between DNA and histone methylation in fungi, and they commonly regulate fungal development and mycotoxin biosynthesis.

## 1. Introduction

Fungal plant pathogens are among the predominant causal agents of plant diseases, and they are responsible for extensive losses in the yield and quality of many economically important agronomical, horticultural, ornamental, and forest plants worldwide [1,2,3]. Specifically, phytopathogens can cause huge losses of total production of vegetables and fruits in industrialized countries and over 50% in developing countries each year [4,5]. Pathogenic fungi utilize diverse strategies to colonize plants and trigger disease [1,6,7,8]. Some fungi kill their host and feed on dead material (necrotrophs), while others colonize living tissue (biotrophs) [7,8,9,10]. Successful invasion of plant hosts requires tight regulation of pathogenic development and formation of specialized infection structures [7,11,12,13,14]. Plenty of fungal effector molecules, which have special functions in plant–pathogen interactions, can not only adapt phytopathogenic fungi to plant physiology, but also govern the virulence of fungal invaders during colonization [15,16]. On the one hand, ambient pH, a common environmental signal, can influence a wide range of biological phenomena by regulating intracellular pH and impairing protein synthesis in phytopathogenic fungi [17]. Correspondingly, many important phytopathogenic fungi [2], such as *Alternaria alternata* [18], *Botrytis cinerea* [12], *Colletotrichum gloeosporioides* [19], *Fusarium oxysporum* [20], *Penicillium expansum* [21], and *Sclerotinia sclerotiorum* [22], adjust the activities of extracellular virulence factors according to different pH signals by regulating their transcription levels. The accumulation of transport vesicles and reduction in the levels of extracellular proteins in the *B. cinerea ΔBcsas1* mutant show that Bcsas1 plays an essential role in the secretory pathway [9]. Many transcription factors (e.g., Bcmads1 in *B. cinerea* [13] and PePacC in *P. expansum* [21]) are involved in the control of growth and virulence in fungal plant pathogens [7]. On the other hand, reactive oxygen species (ROS) have been shown to mediate fungal growth and pathogenicity [11,23,24]. The NADPH oxidase (NOX) complex plays an important role in the response to oxygen stress [7,14], influencing the growth, sporulation, and virulence of *B. cinerea* by regulating the expression of BcPGD [23,24]. The transcription factor BcLTF1 participates in maintaining the balance between production and scavenging of ROS [25]. Interestingly, the lack of Rho3, a small GTP-binding protein, reduces ROS accumulation in the hyphae tips of *B. cinerea* [26]. In addition, aquaporins (AQPs) are ubiquitous water-channel proteins, and deletion of its ortholog, AQP8, in *B. cinerea* results in the inhibition of the development of conidia and infection structures [27]. To further colonize hosts and establish disease, fungal pathogens deploy a plethora of virulence factors (e.g., PeCRT and PeSAT) [6,7,21]. Some virulence factors are upregulated to facilitate effective host colonization and infection [28,29]. For example, the fungal plant pathogen *Mycosphaerella graminicola* strongly upregulates the expression of *Mg3LysM* and *Mg1LysM* genes [30]. Some virulence factors are downregulated to avoid host recognition and dampen host defence responses [31]. The crucial β-1,6-glucan synthesis genes are transcriptionally downregulated in phytopathogenic fungi *Colletotrichum graminicola* during infection [31]. Overall, deciphering fungal pathogenesis not only allows us to better understand how fungal pathogens infect host plants but it also provides valuable information for the control of plant diseases, including new strategies to prevent, delay, or inhibit fungal development. Recently, scientists have begun to focus on the role of epigenetics in the regulation of the growth and pathogenicity of fungal plant pathogens [32].

DNA methylation is a basic and significant epigenetic modification of genomic DNA in eukaryotes [33,34]. Rollin Hotchkiss was the first to identify a modified cytosine in 1948 and inferred that it was 5-methylcytosine (5mC) [35]. The methylation of DNA is normally catalyzed by a conserved set of proteins called DNA methyltransferases (MTases), which can add a methyl group to cytosine, giving rise to 5mC [33,34]. By changing the bond between DNA-binding proteins and DNA sequences, or the recruitment of proteins [6], DNA methylation has significant effects on numerous biological processes, including gene expression, genomic imprinting, transposon silencing, and chromosome stability [33,34,35,36]. DNA methylation generally distributes in gene promoter regions, transposable elements (TEs), repeat sequences, and transcribed regions of genes [37,38]. In mammal genomes, approximately 70−80% of cytosine methylation exists in CG islands [39]. In plant genomes, cytosine methylation distributes in all common sequence contexts, including the symmetric CG and CHG contexts (where H can be A, T, or C) and the asymmetric CHH contexts [33,34,40]. In the *Arabidopsis* genome, the levels of CG, CHG, and CHH methylation are approximately 24%, 6.7%, and 1.7%, respectively [41]. Although DNA methylation levels are very low or barely detectable in most fungi [6,33], transcriptome and methylome analyses have indicated that DNA methylation is related with the silencing of TEs and gene expression in filamentous fungi [6,33,34,35,36].

DNA methylation is relatively stable and heritable and it can also change dynamically during different developmental stages and plays a central role in coordinating developmental programs in mammals, plants, and fungi [41,42,43,44]. For instance, dysregulated methylation, which usually takes place in promoter regions or CG dinucleotides, causes many human illnesses [45]. DNA methylation patterns also change in response to environmental stresses during plant growth and development [38,41,46]. Dynamics of DNA methylation are necessary for known crucial developmental processes, such as sexual reproduction in Arabidopsis and Asian cultivated rice [47,48]. Recent methylome studies have revealed that DNA methylation in fungi exhibits significant variation and exerts important functions in fungi [40]. For example, DNA methylation inhibits transcription elongation in *Magnaporthe oryzae* [44] and the filamentous fungi *Neurospore crassa* [49]. In the latter, approximately 2% of cytosines in the genome are methylated [50].

Although DNA MTases and genome-wide DNA methylation patterns have been identified in many fungal plant pathogens, their functions and mechanisms of action are still poorly understood. DNA methylation is implied in plant–pathogen interactions, and insights into DNA methylation in plant–pathogen interactions might be informative in understanding the molecular basis of pathogenesis and host responses, which can make contribution to controlling plant diseases. Here, we review and discuss the current progress on the features of DNA MTases and methylation patterns in fungi. We also summarize the functions of dynamic methylation in fungal development, pathogenicity, and secondary metabolism.

## 2. DNA Methyltransferases in Fungal Plant Pathogens

DNA methylation is established through complex genetic pathways that depend on both *de novo* and maintenance DNA MTases that transfer an activated methyl group from S-adenosyl-L-methionine to the C^5^ position of the cytosine ring [33,34]. In summary, DNA MTases in eukaryotes belong to five groups on the basis of their structure and function (Figure 1) [41,51]: (1) the maintenance MTase family, which contains the animal DNMT1 [39], plant methyltransferase 1 (MET1) [52], and fungal DIM-2 [53]; (2) the de novo MTase family, which contains the animal DNMT3A and DNMT3B [39] and plant domains rearranged methyltransferase (DRM2) [54]; (3) the plant-specific flowering maintenance chromomethylase (CMT) family, which contains the *Arabidopsis thaliana* CMT2 and CMT3 [55]; (4) the fungal-specific MTase-like family, which contains the *Ascobolus immersus* Masc1 [56] and *N. crassa* RID (*RIP D*efective) proteins [57]; and (5) the predicted CpG-specific maintenance MTase family, which contains the DNMT5 [58]. DNA MTases of the first three families have been proved to methylate cytosines in vitro, while no such activity has been directly demonstrated for the other two families [56,57,58]. In addition, a sixth family contains tRNA methyltransferases, which are typified by DNMT2 and can methylate tRNAs particularly at C38, protecting tRNAs from breaking by ribonuclease angiogenin [59].

DNA methyltransferases are indispensable for the normal development of most eukaryotes [52,53,54]. These enzymes function within a classical regulatory mechanism and they are widespread in fungi to animals, playing multifarious roles by repressing the gene expression and transposons [39,60,61]. While there are a few reports on the study of fungal DNA methylation, some putative DNA MTases have been found in fungal species owning a pretty low abundance of 5mC (e.g., *Aspergillus nidulans* [58]). These MTase-like proteins possess an essential function as well.

The number and kind of DNA MTases vary greatly between different fungi [44,57,58]. Phylogenetic analysis of over 500 fungal species and strains showed that fungal 5mC MTases are predominantly divided into two monophyletic clades: (1) the DNMT1 clade, which contains DIM-2, DNMT1, DNMT5, and RID, and (2) the DNMT2 clade, which includes tRNA^Asp^ methyltransferases [61]. Generally, ascomycete fungi have a DNMT1-related DNA methyltransferase, DIM-2, which can methylate repeat DNA sequences and TEs [53]. On account of the DNA MTase differentiation, which happened in the ancestors of all living eukaryotes, DIM-2 and RID are derived in fungi, with RID evolving earlier than DIM-2 [37,58]. Recently, no DNA MTases are found in several subphyla, including Saccharomycotina, Cryptomycota, and Microsporidia [58,62]. In contrast, the collocation of DNA MTases is diverse in Ascomycota, Basidiomycota, Mucoromyceta, and Zoopagomycota [58,63]. The top three most common combinations of DNA MTases in fungi are DNMT1 + DNMT5, DIM-2 + DNMT5 + RID, and DNMT1 [58]. Only a few ascomycete fungi, such as *B. cinerea* and *Pseudogymnoascus destructans*, possess all kinds of DNA MTases (DIM-2 + DNMT1 + DNMT5 + RID) [58].

DNA MTase homologs have been identified in many fungal pathogens, including *Arthrinium arundinis*, *A. fumigatus*, *B. cinerea*, *Caliciopsis orientalis*, *Fusarium graminearum*, and *M. oryzae* [58]. However, only a few of them have been verified to have methyltransferase activity [6,63]. Two putative DNMT1 homologs, Masc1 in *A. immerses* and RID in *N. crassa*, have been demonstrated to function in repeat-induced point (RIP) mutation, and methylation induced premeiotically (MIP) mutation, respectively [63,64]. In *Ascobolus*, Masc2 possesses DNA MTase activity in vitro; nevertheless, knockout of *masc2* gene have no effect on MIP mutation or methylation patterns [65]. AnDmtA and AlDmtA are identified as DNA MTase homologs and are essential for sexual development in *A. nidulans* and activation of the aflatoxin (AF) biosynthesis gene cluster in *A. flavus*, respectively [63,66]. However, the levels of DNA methylation in *Aspergillus*, and especially in *A. flavus*, are extremely low [66]. The numbers of 5mC sites in wild type (WT), *ΔModim-2*, and *ΔMorid* are respectively 46,124, 4563, and 36,809, which proves that MoDIM-2 is a major DNA MTase in *M. oryzae* [44]. Furthermore, approximately a quarter of 5mC sites in Δ*Morid* do not overlap with those in WT, which shows that MoRID has an important function in regulating methylation specificity [44]. However, no sequence preference is observed for MoRID-dependent cytosine methylation in *M. oryzae*, which is different from that in *N. crassa* [44]. DNA methylation levels were 71%, 10%, and 8% that of WT in *ΔMrRID*, *ΔMrDIM-2*, and *ΔMrRID/DIM-2*, respectively, showing that deletion of both MrRID and MrDIM-2 exerts an additive effect on DNA methylation in *Metarhizium robertsii* [67]. Although all members of the DNMT1 family have been found in fungi, no homologs of DNMT3 have been identified in any fungal species to date (Figure 1) [44,58].

## 3. Patterns of DNA Methylation in Fungal Plant Pathogens

The majority of DNA methylation sites are distributed in TEs, gene promoter regions, and the repetitive DNA sequences, and the DNA methylation patterns are dynamically changing and inheritable, responding to physiological conditions and environmental stimuli (Figure 2) [44,50,68,69]. Recent genome-wide analysis of DNA methylation has revealed that methylation levels are low in fungi, ranging from below the detection threshold to just above the detection threshold for both CpG site methylation and non-CpG methylation [45,63]. Approximately 1.5% of cytosines are methylated in *N. crassa*, while < 0.1% of cytosines are methylated in *Schizosaccharomyces pombe* and *A. nidulans* [61,63,70]. Moreover, 5mC levels are on average higher in Basidiomycota than in other phyla regardless of genomic location and sequence context [58]. Methylated cytosines are also preferentially found in CpG dinucleotides of repetitive DNA sequences and TEs across fungal genomes [44,71]. Global methylation levels in the range of 3.3–5.2% were observed in *Heterobasidion parviporum* with a pronounced preference for CpG dinucleotides (6.7–9.3%) over nonCpG nucleotide contexts (2.0–3.7%), which is a phenomenon also shared by most species for which methylation patterns have been studied [37,71]. In *H. parviporum*, the relative proportion of 5mC sites in the three nucleotide contexts is similar (47–52% in CHH, 30–36% in CpG, and 17–18% in CHG), indicating a methylation homogeneity in terms of the three types of nucleotide contexts [71]. Importantly, DNA methylation in different genomic regions may differentially influence gene activity depending on the underlying sequence. In the following sections, we further discuss the distribution of DNA methylation in different genomic regions.

An increasing number of studies have reported the presence of cytosine methylation in the gene bodies of many eukaryotic organisms, including fungi [72]. The gene body is considered to begin after the first exon because methylation of the first exon, like promoter methylation, leads to gene silencing [73]. There is some evidence to suggest that moderately expressed genes are likely to be methylated in the gene body in animals and plants [37]. In contrast, CG-enriched genes in fungi do not exhibit the same normal-like distribution of CG methylation across the gene body as plants and some insect species [58,74]. The methylation levels of transcribed genes in the CpG context are lower than in their flanking regions in *H. parviporum* [71]. In addition to methylation in TEs, approximately 20% of nonTE genes are also methylated in the mycelia of *M. oryzae* [44]. In gene bodies of *M. oryzae*, 5mC is frequently found near the start and end of coding regions and distant from the center [44]. Within the genic regions in *Cryphonectria parasitica*, exons present the highest proportion (28–45%) of strain-specific 5mC sites [75].

According to the methylation data from more than 40 fungal species, most methylated cytosine bases exist in transcriptionally silent and repetitive loci, and they are absent in transcriptionally activated genes, indicating that fungal DNA methylation preferentially contributes to TE suppression for maintaining the genome integrity [58]. Analysis of genomic DNA methylation patterns confirmed that TEs are heavily methylated in both CpG (>90%) and nonCpG (>20%) sites in different developmental stages, and only a few TEs are expressed in *H. parviporum* [71]. Analogously, 5mC sites in mycelia are not evenly distributed but clustered across chromosomes, forming densely methylated domains around TE-rich and gene-poor regions [6,44]. When transcription of TEs was compared between WT and *ΔModim-2 M. oryzae* strains, changes in TE transcript abundance varied in a manner that was dependent on type and genomic location and not on the presence of DNA methylation [44]. Interestingly, in fungi, a marked trend toward hypomethylation is observed for TEs located within 1-kb of expressed genes, rather than segregated in TE-rich regions of the fungal genome [69].

DNA methylation levels vary according to cell type in gametophytes and change dynamically during eukaryotic growth and development [36,44,71]. In *Cordyceps militaris*, the methylome undergoes global reprogramming during development, such that pre-existing 5mC sites are demethylated while C sites are methylated at different loci [74]. Variation in the methylome of *H. parviporum* is observed during its asexual development and different lifestyles, reinforcing the dynamic nature of DNA methylation [72]. In *M. oryzae*, methylation peaks in the regions flanking coding sequences in mycelia and disappears in conidia and appressoria [44], suggesting that DNA methylation patterns change according to developmental stage.

## 4. The Function of DNA Methylation in Fungal Plant Pathogens

### 4.1. DNA Methylation and RIP Mutation

DNA methylation in fungi is strongly associated with sequences influenced by RIP mutation, which is a gene silencing mechanism and a primary, fungal-specific genomic defense system which is closely associated with DNA methylation during the sexual cycle (Figure 3) [50,57]. Selker and colleagues defined RIP as a biological process that generally happens in haploid parental nuclei after fertilization [50]. Following its first discovery in *N. crassa*, RIP has been experimentally confirmed to exist in numerous ascomycete species, such as *M. oryzae*, *Podospora anserina*, *Leptosphaeria maculans*, and *Fusarium graminearum* [76]. Further reports revealed that the RIP pathway introduces cytosine to thymine (C to T) transitions, which decreases GC-content and generally accompanies cytosine methylation [57,77]. RIP mechanism serves to recognize chromosomal DNA sequences longer than 400 bp and is not impacted by their transcriptional states and locations in the genome [76,77].

RIP mutation can be regulated by two special molecular pathways in *N. crassa* [57,77,78]. The first pathway, which is canonical and regulated by RID, mainly results in the repeated sequence mutations [57]. The second pathway, which is mediated by DIM-2, preferentially results in mutations in the single-copy linker region between repeated DNA sequences, and normally takes charge of cytosine methylation in vegetative cells of *N. crassa* [53,78,79,80]. Apart from DIM-2, the second pathway also needs some special proteins with certain functions in heterochromatin formation, such as a SUV39-lysine-methyltransferase (known as DIM-5 for short) [80]. Substantial evidences proved the existence of RIP mutation or a RIP mutation-like process in *A. fumigatus*, *F. graminearum*, and *M. grisea* [80,81,82] and the effects of repeated sequences on DNA methylation status by RIP mutation in an indirect regulating means [57,77]. RIP mutation primarily happens at (A/Tp)Cp(A/T) contexts and is associated with cytosine methylation during the fungal sexual stages [80]. There is no further proof to show how maintenance or de novo methylation affects the DNA methylation patterns of fungal phytopathogens via RIP mutation in detail.

### 4.2. Impact of DNA Methylation on the Development of Fungal Plant Pathogens

Genomic DNA methylation in eukaryotic organisms is tightly linked to the modulation of transcriptional silence, genome integrity, genomic imprinting, and other processes [35,40,44]. The methylation of transcriptional regions, or gene promoters in plants and mammalians is strongly associated with the control of antisense transcription and transcriptional elongation [40,58,73,83]. In contrast, DNA methylation in fungi usually is regarded as a genomic defense mechanism, because fungal DNA methylation primarily spreads over TEs, repeated sequences, and heterochromatic regions, leading to silencing these genomic regions and affecting the development process [37,44,57]. Apart from its functions in fungal genomic defense [44,71,75], DNA methylation is confirmed to make essential contributions to development in *M. oryzae* [44], regulation of secondary metabolism in *Ganoderma sinense* [76], and morphogenetic change in *C. parasitica* [75] by regulating the transcription of relevant genes. DNA methylation changes dynamically during the asexual stages of *M. oryzae* [44]. Methylome analysis showed that the DNA methylation loci in conidia and appressoria of *M. oryzae* are relatively fewer than in mycelia [44]. Mutation of *CpBck1* in *C. parasitica* leads to the significant changes of the DNA methylation proportion and a sporadic occurrence of sectors [75]. DNA methylation impacts fungal development by regulating the transcription of genes related to metabolic activities and energetic metabolism in the *M. robertsii* [67]. On the contrary, DNA methylation is not directly associated with gene expression involved in sexual development in *C. militaris* [74]. Notably, the function of DNA methylation in fungal plant pathogens is complicated and divergent, so further work should investigate its regulatory mechanisms during fungal development.

### 4.3. Effect of DNA Methylation on Fungal Pathogenicity

Although numerous fungi have no or low DNA methylation, the roles of DNA methylation in controlling the pathogenicity of pathogenic fungi has been demonstrated. The spore median lethal times (LT50s) for the *ΔMrDIM-2* and *ΔMrRID/ΔDIM-2* strains in *Galleria mellonella* were respectively decreased by 47.7% and 65.9%, proving that MrDIM-2 is crucial for the pathogenicity of *M. robertsii* [67]. DMT1, which is a DNA MTase ortholog of Dim-2, contributes to the changes of MAGGY methylation status between *M. oryzae* isolates, Br48, and GFSI1-7-2 [63]. While MoDMT1 has no obvious role in governing the development and pathogenicity of *M. oryzae* [63], MgDIM-2 in *M. grisea* plays an important and significant role in conidiation and appressorium differentiation. Nevertheless, *ΔMoDIM-2* did not present the distinct change of virulence [84]. Some research groups have considered that DNA methylation indeed exists in *Aspergilli* species, and about 0.25% of cytosines are methylated [85]. Treatment with the methylation inhibitor leads to morphological changes [86], and virulence in *ΔAfdmtA* is altered [87]. *ΔAfdmtA* strains can infect peanut seeds and maize kernel and develop more rapidly on crop seeds than WT, proving that the DNA methyltransferase maybe has an important role in pathogenicity [66]. Apart from the phytopathogenic fungi, DNA methylation has been implicated in the regulation of plant–fungi interactions and it plays a great role in disease resistance and defense priming of host plants [88]. Studies prove that DNA methylation has a positive effect on the expression of disease-resistance genes, such as *Pib*, which has hypermethylation at its promoter and can be induced by *M. grisea* [88].

### 4.4. Association between DNA Methylation and Secondary Metabolism

Genes that are involved in secondary metabolism are sequentially distributed in biosynthetic gene clusters (BGCs), providing the coordinated regulation of the genes that are related to any metabolite [89]. The treatment with DNA methylation inhibitors, such as 5-azacytidine (5-AC) and RG-108, which are commonly used to inhibit the activity of DNA MTases, have been successfully confirmed to induce the expression of silent BGCs and influence secondary metabolites in *Alternaria* species [90], *A. niger* [91], *Cladosporium cladosporioides* [92], *Diatrype* sp. [92], *Penicillium citreonigrum* [93], and *Isaria tenuipes* [94]. Among *Aspergillus* species, *A. flavus* is the most influential and important, and it can synthesize carcinogenic secondary metabolites, aflatoxins (AFs), which are a special family of oxygenated polyketide-derived toxins [86,95]. Treatment with 5-AC results in production reduction of AFs in *A. flavus* [86]. Compared with WT, *ΔAfdmtA* strains present decreased transcriptional level of genes in the AF cluster, leading to the reduction of conidiation and AF biosynthesis, supporting the finding that DNA methylation plays an important role in AF metabolism [66]. In addition, the combined action of 5-AC and methotrexate resulted in a 30% increase in light-induced carotenoid synthesis, which were highly toxic for *N. crassa* growth [96].

## 5. DNA Methylation and Histone Methylation in Fungi

Recently, it has been demonstrated that there is a biological relationship between DNA and histone methylation, which both are correlated with different chemical reactions. This relationship has a vital function in gene silencing from fungi to mammals [96,97,98,99,100]. The histone modification markers, such as H3K4me3, may serve as indirect regulators to influence DNA methylation [61,101]. PRMT5 is the primary methyltransferase in charge of histone arginine methylation (H3R8me), and the decreased levels of PRMT5 can diminish the binding between DNMT3A and chromatin, reduce DNA methylation, and subsequently facilitate genetic transcription [98]. Current evidence suggests that DNA and histone methylation commonly regulate fungal development and mycotoxin biosynthesis [99]. Histone methylation is readily reversible and generally precedes DNA methylation in *N. crassa*, while DNA methylation is relatively stable and conduces to form a stable heterochromatic state [97,100]. DIM-2 in *N. crassa* is associated with DNA methylation and the silence of gene transcriptions [53] when DIM-5 takes charge of trimethylation of H3K9, which facilitates the methylation of nearby DNA loci [101]. Abundant research data show that some special proteins, which can recognize methylated DNA and methylated histones, are essential for recruiting methyltransferases to correct genomic sites in fungi [100]. For example, the H3K9me3 marker is established by the DCDC histone methyltransferase complex, including DIM-5, DIM-7, DIM-9, CUL4, and DDB1 [102,103], which is targeted to AT-rich DNA and typical of RIP [102]. After H3K9me3 is generated by DIM-5, heterochromatin protein 1 can accurately identify the H3K9me3 marker and contributes to recruiting DIM-2 to catalytic methylation at the same locus [100].

## 6. Conclusions

DNA methylation is proven to be engaged in the persistence of fungal plant pathogens in plant hosts and is at the forefront of deciphering the interaction between fungal plant pathogens and hosts. DNA methylation is primarily related to the silencing of TEs and repeated DNA [44], which contributes to the development [44,57], pathogenicity [6,63,65,84], and secondary metabolism [64] of fungal plant pathogens. During the fungal development, DNA methylation is reversible and dynamic, responding to environmental and physiological conditions [44]. Furthermore, DNA methylation may help fungal plant pathogens to avoid host defence mechanisms [104] and are strongly involved in host–pathogen interactions [105]. In consideration of the annual losses of fruits caused by phytopathogenic fungi and the importance of epigenetics [3,32], it is very meaningful to figure out the significant roles of DNA methylation in fruit–pathogen interactions. Comparing the dynamic of DNA methylation under different growth and infection stages may highlight the function of DNA methylation on modifying the infection strategies of fungal plant pathogens in response to hosts. However, there is still not enough evidence to clarify the direct relationship between DNA methylation and gene expression in fungi [58], indicating that substantial further research is necessary to investigate DNA methylome of phytopathogenic fungi. To explore the pattern and function of DNA methylation in fungal plant pathogens may be highly informative in exposing how DNA methylation help pathogens colonize hosts and cause disease.

## Figures and Tables

**Figure 1 microorganisms-08-00227-f001:**
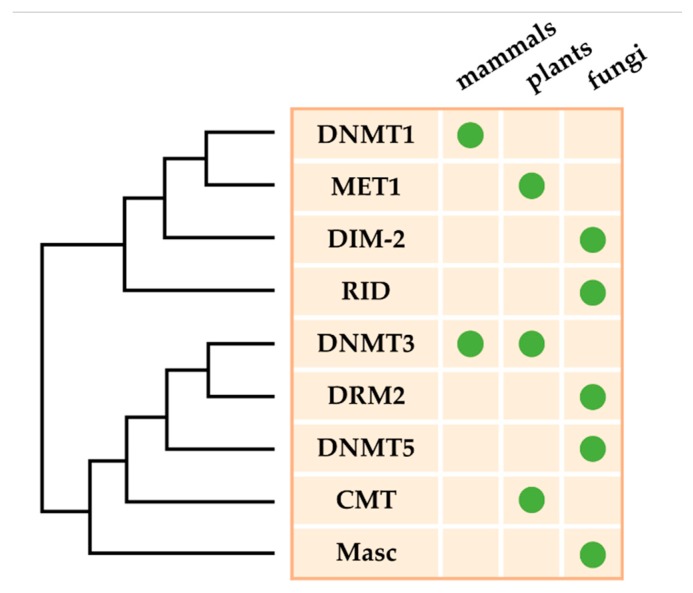
Evolutionary relationship of eukaryotic DNA MTases. Although DNMT1 homologs are found in almost all eukaryotes that exist in DNA methylation, lineage-specific losses and gains of DNA MTases are found in specific taxa. This phylogeny is a representation and it is not applicable to all species within each lineage owing to recurrent loss of the DNA methylation machinery.

**Figure 2 microorganisms-08-00227-f002:**
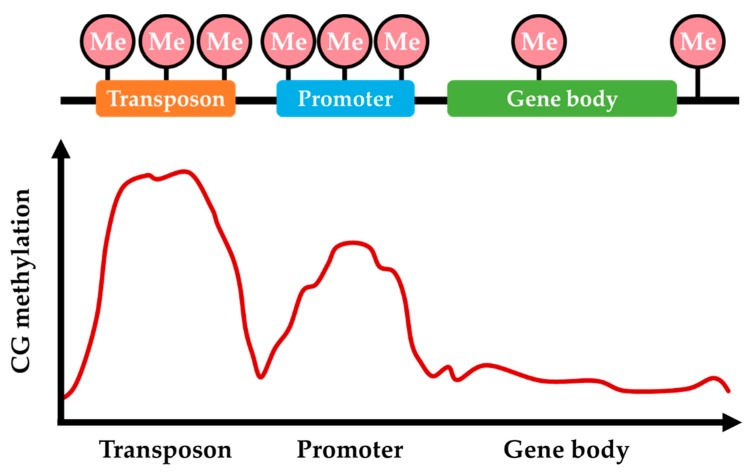
The pattern of DNA methylation in different regions of the fungal genome. Cytosine methylation preferentially distributes in transposons and prompter regions and rarely distributes in gene body and intergenic regions.

**Figure 3 microorganisms-08-00227-f003:**
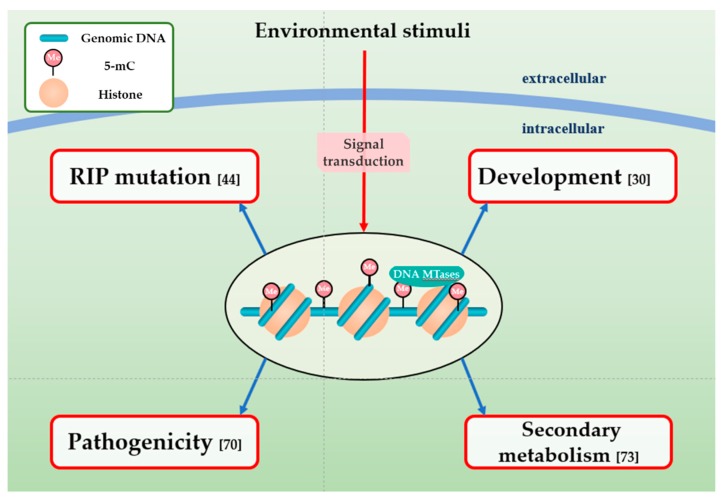
Functions of DNA methylation in fungal pathogens. DNA methylation in fungal plant pathogens can respond to environmental stimuli and involved in many biological processes, including RIP mutation, development, pathogenicity, and secondary metabolism.

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
