# Peer review of "The Pattern and Function of DNA Methylation in Fungal Plant Pathogens"

_microorganisms, 2020, doi:10.3390/microorganisms8020227_

Round 1

Reviewer 1 Report

Line   Comment

22 and they are

46 but it also provides

55 DNA methylation has

62 Arabidopsis

68 and it plays a central role

102 and it is not applicable

105 and they are widespread in fungi

112 divided into two

116 which happened

122 possess all kinds

136 Besides, approximately a quarter

147 inheritable, responding

180 methylation data from more than 40

181 and they are absent

217 can respond to  

229 There is no

254 has been demonstrated

260 important and significant

268 Genes which are involved

277 reduction of AFs

279 supporting the finding that

283 Recently it has been demonstrated

284 This relationship 

292 while DNA methylation

295 Abundant research data

306 dynamic, responding to

307 to avoid host defence

308 there is still not enough

310 research is necessary

312 in exposing how DNA

Author Response

Response to Comments of Reviewer 1

We received your comments and suggestions about our submitted manuscript entitled “The pattern and function of DNA methylation in fungal plant pathogens” (Manuscript ID: microorganisms-703781). Above all, we would like to express our gratitude to you for giving us such helpful advice about revising this manuscript again. According to the suggestions, we have earnestly revised this manuscript by using the "Track Changes" function in Microsoft Word. All the changes, in detail, made during revision are as follows:

Point 1: 22 and they are

Response 1: We have added “they” in line 27 (in revisions mode).

Point 2: 46 but it also provides

Response 2: We have added “it” in line 55.

Point 3: 55 DNA methylation has

Response 3: We have corrected “have” to “has” in line 64.

Point 4: 62 Arabidopsis

Response 4: We have set “Arabidopsis” in italics in line 71.

Point 5: 68 and it plays a central role

Response 5: We have changed “and plays a critical role” to “and it plays a central role” in line 77.

Point 6: 102 and it is not applicable

Response 6: We have added “it” in line 111.

Point 7: 105 and they are widespread in fungi

Response 7: We have changed “is widespread fungi to animals” to “and they are widespread in fungi to animals” in line 114 to 115.

Point 8: 112 divided into two

Response 8: We have corrected “to” to “into” in line 122.

Point 9: 116 which happened

Response 9: We have added “which” in line 126.

Point 10: 122 possess all kinds

Response 10: We have replaced “own all the kinds” with “possess all kinds” in line 132.

Point 11: 136 Besides, approximately a quarter

Response 11: We have added “a” in line 146.

Point 12: 147 inheritable, responding

Response 12: We have deleted “as” in line 157.

Point 13: 180 methylation data from more than 40

Response 13: We have changed “the methylome which are belonged to” to “the methylation data from” in line 191.

Point 14: 181 and they are absent

Response 14: We have added “they” in line 192.

Point 15: 217 can respond to 

Response 15: We have corrected “can be responded” to “can respond” in line 228.

Point 16: 229 There is no

Response 16: We have changed “Nowadays, there” to “There” in line 241.

Point 17: 254 has been demonstrated

Response 17: We have changed “testified” to “demonstrated” in line 266.

Point 18: 260 important and significant

Response 18: We have added “and” in line 272.

Point 19: 268 Genes which are involved

Response 19: We have corrected “Genes which involve” to “Genes which are involved” in line 284.

Point 20: 277 reduction of AFs

Response 20: We have replaced “drawdown” with “reduction” in line 293.

Point 21: 279 supporting the finding that

Response 21: We have added “the finding” in line 295.

Point 22: 283 Recently it has been demonstrated

Response 22: We have change “certain” to “demonstrated” in line 300.

Point 23: 284 This relationship

Response 23: We have changed “and this” to “This” in line 301.

Point 24: 292 while DNA methylation

Response 24: We have corrected “meanwhile” to “while” in line 309.

Point 25: 295 Abundant research data

Response 25: We have replaced “proofs” with “data” in line 313.

Point 26: 306 dynamic, responding to

Response 26: We have replaced “dynamically changes” with “dynamic” in line 324.

Point 27: 307 to avoid host defence

Response 27: We have corrected “avoiding host defense” to “avoid host defence” in line 325 to 326.

Point 28: 308 there is still not enough

Response 28: We have changed “no” to “not” in line 327.

Point 29: 310 research is necessary

Response 29: We have corrected “researches are” to “research is” in line 328.

Point 30: 312 in exposing how DNA

Response 30: We have replaced “expounding” with “exposing” in line 330.

Reviewer 2 Report

The paper entitled “The pattern and function of DNA methylation in fungal plant pathogens” reports an interesting view about a key topic for plant pathogens. It is a very well written paper, and minor revision are requested before publication in Microorganisms.

Abstract

The abstract is a bit generic and poorly informative. I suggest to include in the abstract some key sentences about some of the main mechanisms reported in the review.

Text

L.22-24. Fungal are responsible even to dangerous diseases for not food-related plants (eg. ornamental plants, forests), so I suggest to improve the sentence and literature. Otherwise, if the review is limited to food-plant related pathogens, please state this approach.

L.24-26. This sentence is even more specific, limiting the estimation of damage by fungi to post-harvest diseases. Please add details to other sectors or justify your choice.

L.32-45. Please state why Authors chose (mainly) B. cinera for this example.

Fig 2. The figure lacks in x-axes tag: if the tag is the upper scheme, please find a better method of show.

L.203. Please explain RIP acronym.

L.232-281. How many fungal plant pathogens (at least in terms of genera) were investigated so far? The data did not well report if the literature is limited and/or not related to general behavior. Plant-host combinations should be mentioned.

L.303. It is not clear if the knowledge limits are related to the low number of research and the derivable generalization to most of plant pathogens or if pathways are not yet fully described.

Author Response

Response to Comments of Reviewer 2

We received your comments and suggestions about our submitted manuscript entitled “The pattern and function of DNA methylation in fungal plant pathogens” (Manuscript ID: microorganisms-703781). Above all, we would like to express our gratitude to you for giving us such helpful advice about revising this manuscript again. According to the suggestions, we have earnestly revised this manuscript by using the "Track Changes" function in Microsoft Word. All the changes, in detail, made during revision are as follows:

Point 1: The abstract is a bit generic and poorly informative. I suggest to include in the abstract some key sentences about some of the main mechanisms reported in the review.

Response 1: Thanks for your advice. We have improved the abstract as follow (changes are bold and underlined):

To successfully infect plants and trigger disease, fungal plant pathogens use various strategies that are dependent on characteristics of their biology and genomes. Although pathogenic fungi are different from animals and plants in the genomic heritability, sequence feature, and epigenetic modification, an increasing number of phytopathogenic fungi have been demonstrated to share DNA methyltransferases (MTases) responsible for DNA methylation with animals and plants. Fungal plant pathogens predominantly possess 4 types of DNA MTase homologs, including DIM-2, DNMT1, DNMT5, and RID. Numerous studies have indicated that DNA methylation in phytopathogenic fungi mainly distributes in transposable elements (TEs), gene promoter regions and the repetitive DNA sequences. As an important and heritable epigenetic modification, DNA methylation is associated with silencing of gene expression and transposon, and it is responsible for a wide range of biological phenomena in fungi. This review highlights the relevant reports and insights into the important roles of DNA methylation in the modulation of development, pathogenicity, and secondary metabolism of fungal plant pathogens. Recent evidences prove that there are massive links between DNA and histone methylation in fungi, and they commonly regulate fungal development and mycotoxin biosynthesis.

Point 2: L.22-24. Fungal are responsible even to dangerous diseases for not food-related plants (eg. ornamental plants, forests), so I suggest to improve the sentence and literature. Otherwise, if the review is limited to food-plant related pathogens, please state this approach.

Response 2: Thanks for your suggestion. We have widened the range of plant diseases caused by fungal plant pathogens and improved this sentence as follow (in line 28 to 30):

Fungal plant pathogens are among the predominant causal agents of plant diseases, and they are responsible for extensive losses in the yield and quality of many economically important agronomical, horticultural, ornamental, and forest plants worldwide [1,2,3].

In addition, we have added the new reference in line 348 to 350.

Point 3: L.24-26. This sentence is even more specific, limiting the estimation of damage by fungi to post-harvest diseases. Please add details to other sectors or justify your choice.

Response 3: We are pleased to accept your suggestion. We have rechecked the references and revised this sentence as follow (in line 32 to 33):

Specifically, phytopathogens can cause huge losses of total production of vegetables and fruits in industrialized and developing countries each year [4,5].

In addition, we have added the new reference in line 354 to 355, and changed the serial numbers of the other references.

Point 4: L.32-45. Please state why Authors chose (mainly) B. cinerea for this example.

Response 4:  Botrytis cinerea, a typical necrotrophic fungus, is the second most important phytopathogenic fungus around the world, and it can cause grey mould disease in more than 200 crop species worldwide [2]. B. cinerea is now an important model organism of fungal plant pathogens, and has been focused and studied by many research teams for a long time. It is well worthy of clarifying the pathogenic mechanism of B. cinerea, which can help researchers understand the universal pathogenesis and regulatory mechanisms of fungal plant pathogens. We have improved the sentence as follow (in line 40 to 42):

Correspondingly, Botrytis cinerea and Penicillium expansum, two important phytopathogenic fungi around the world, adjust the levels of secreted proteins according to different pH signals by regulating their transcription levels [2,11,15].

Point 5: Fig 2. The figure lacks in x-axes tag: if the tag is the upper scheme, please find a better method of show.

Response 5:  Thanks for your suggestion. We have modified the figure 2 and replaced it in line 174.

Point 6: L.203. Please explain RIP acronym.

Response 6:  Thank you for your question. The RIP acronym stands for repeat-induced point, and we have noted the full term of RIP acronym when this acronym appeared firstly in line 138.

Point 7: L.232-281. How many fungal plant pathogens (at least in terms of genera) were investigated so far? The data did not well report if the literature is limited and/or not related to general behavior. Plant-host combinations should be mentioned.

Response 7:  Until now, there are at least 10 fungal plant pathogens which have been investigated for DNA methylation. The phylogenetic analysis of 528 fungi species/strains reveals that many fungal plant pathogens possess potential DNA MTases, and the WGBS data shows DNA methylation levels are very low or barely detectable in 10 phytopathogenic fungi, including Aspergillus flavus, Botrytis cinerea, Fusarium fujikuroi, Heterobasidion irregulare, Leptosphaeria maculans, Magnaporthe oryzae, Microbotryum lychnidis, Mixia osmundae, Parasitella parasitica, and Wolfiporia cocos [48]. However, DNA methylation has been deeply investigated in only a few fungal plant pathogens, such as Magnaporthe sp. [32], Aspergillus sp. [55], Heterobasidion sp. [58], Cryphonectria sp. [62], Leptosphaeria sp. [63], and Fusarium sp. [63].

According to your advice, we have added the content of plant-host combinations as follow (in line 278 to 282):

Beside the phytopathogenic fungi, DNA methylation has been implicated in the regulation of plant-fungi interaction, and it plays a great role in disease resistance and defence priming of host plants [76]. Studies prove that DNA methylation has a positive effect on the expression of disease-resistance genes, such as Pib, which has hypermethylation at its promoter and can be induced by M. grisea [76].

In addition, we have added the new reference in line 559 to 560.

Point 8: L.303. It is not clear if the knowledge limits are related to the low number of research and the derivable generalization to most of plant pathogens or if pathways are not yet fully described.

Response 8:  On the one hand, the pattern and function of DNA methylation vary greatly among different fungi, and it is hard to reveal abundant and universal knowledge. DNA methylation levels are very low or barely detectable in phytopathogenic fungi, which adds to the research difficulties. On the other hand, there are only a few studies of DNA methylation in fungal plant pathogens recently. Although the contents of conclusions are limited because of finite research findings, this review is the first one to summarize the research progress on DNA methylation of fungal plant pathogens in as much detail as possible.